# A CRITICAL STUDY OF WHAT PRE-TRAINED CODE MODELS (DO NOT) LEARN

## ABSTRACT

Parallel to the recent success of self-attention-based language models across a range of coding assistance tasks, several studies have underscored that pre-trained code models (PCMs) utilize self-attention and hidden representations to encode relations among input tokens. Our research extends upon these insights by understanding the properties of code that PCMs may not fully encode and by broadening the scope to encompass data flow relations. Our study reveals that while PCMs do encode syntactic and data flow relations in self-attention, they only encode relations within specific subsets of input tokens. Specifically, by categorizing input tokens into syntactic tokens and identifiers, we find that models encode relations among syntactic tokens and among identifiers but fail to encode relations between syntactic tokens and identifiers. We show that this limitation results in hidden representations not encoding enough information about input tokens to discriminate between different identifier types and syntax structures. Importantly, we observe that this learning gap persists across different model architectures, datasets, and pre-training objectives. Our findings shed light on why PCMs fail to generalize beyond dataset they are trained on and in real world applications.

## 1 INTRODUCTION

Pre-trained code models (PCMs) are Transformer models (Vaswani et al., 2017) pre-trained on a large corpus of source code and natural language - programming language (NL-PL) pairs. They have become a popular method for coding assistance tasks, including next-token prediction, code completion, code generation from natural language prompts, and program repair (Xu & Zhu, 2022).

However, due to the black-box nature of neural networks, it can be challenging to understand what information PCMs use for prediction and generation. To this end, prior studies have attempted to understand and explain the functioning of PCMs. Some of these studies argue that models can learn syntactic and semantic structure of code (Wan et al., 2022; Troshin & Chirkova, 2022; López et al., 2022) and do understand code logic (Baltaji & Thakkar, 2023). Other studies suggest that models do not generalize well (Hajipour et al., 2022; Hellendoorn et al., 2019), learn shortcuts (Sontakke et al., 2022; Rabin et al., 2021), and memorize training inputs (Rabin et al., 2023a; Yang et al., 2023b). These diverging views play out in practice. Despite being successful on various downstream tasks, the output generated by PCMs has compilation errors, due to syntactical mistakes (Le et al., 2022), as well as semantic errors like random identifiers (Guo et al., 2021), and can invoke undefined or out-of-scope functions, variables and attributes (Chen et al., 2021).

This paper contributes new insights about the question what PCMs learn and do not learn. We conduct a fine-grained analysis of self-attention and hidden representation of PCMs at code token level, which shows that while PCMs encode some code relations well, they also miss out on some important ones, which limits their ability to understand syntax and code logic.

There are different types of relations between code tokens, including relations in an abstract syntax tree (AST), as well as, data flow or control flow relations between code blocks. Similar to Wan et al. (2022), we focus on syntactic relations in the AST and create a syntax graph with edges between code tokens within a motif structure (Figure 1b). Since, such a syntax graph does not encompass all the relations among identifiers, in particular how values flow from one variable to another, we also create a data-flow graph with edges among related variables following Guo et al. (2021).

Previous studies examining the code comprehension ability of PCMs have analyzed all input tokens together, without distinguishing between different categories of code tokens such as identifiers (e.g. function names, function arguments, variables) and syntactic tokens (e.g. keywords, operators, parentheses). However, aggregate studies over all tokens can potentially hide information about what PCMs do not encode. To investigate whether there are specific relations that PCMs fail to encode, we analyze the syntactic-syntactic, identifier-identifier, and syntactic-identifier relations that are encoded in the self-attention values and hidden representations separately.

We analyze the ability of PCMs to understand relations between code tokens in two ways. First, we study how well relations are encoded in self-attention values. Second, we use three prediction tasks to study whether hidden representations encode information about code relations. The first two tasks are for data-flow edges and siblings in the AST, respectively. They enable to investigate whether the hidden representations encode the data flow relations among identifiers, respectively the syntactic relations among tokens. The third task is a tree distance prediction and enables studying whether the encoded information is sufficient to understand subtle differences in code syntax.

Attention analysis (Wan et al., 2022) and probing on hidden representation (Belinkov, 2022) have been used in the past to study what PCMs encode. But prior works relied on non-systematically validated assumptions. We examine these assumptions and assess their impact on the conclusions drawn. For attention analysis, we examine the influence of the attention threshold and the evaluation metric, an analysis which has not been conducted previously (Wan et al., 2022; Vig et al., 2021). For probing, we explore whether the code relations among tokens are encoded linearly or non-linearly. Such an exploration has not been performed so far and prior works assume a linear encoding of syntactic and semantic relations. However, previous work in NLP White et al. (2021) has shown that hidden representations of language models encode some linguistic properties non-linearly.

Our study considers different transformer architectures (encoder-only and encoder-decoder), pre-training objectives, and training datasets. In summary, the results are as follows:

- We provide evidence that prior work often made incorrect assumptions in their experimental settings, e.g., we find that hidden representations encode syntactic relations non-linearly.
- The attention maps of PCMs fall short in encoding syntactic-identifier relations, while they do encode syntactic-syntactic and identifier-identifier relations. For instance, in Figure 1, the keyword `if` attends to itself, `is`, and `:` but not to the related identifier `ignore`.
- Hidden representations do not encode sufficient information to discriminate between different identifier types and to understand subtle syntactical differences. For instance, while the information is sufficient to understand which identifiers are siblings of the keyword `def`, it is insufficient to understand which siblings are function names or (default) parameters.

Our comprehensive analysis provides valuable insights that should inspire further research to devise methods to address current models' limitations.

## 2 BACKGROUND

**Attention Analysis.** In NLP, attention analysis investigates whether self-attention corresponds to linguistic relations among input tokens. For PCMs, attention analysis quantifies how well self-attention encodes relations among code tokens, such as relations in an AST.

**Probing on Hidden Representation** is a technique to study the properties encoded in the hidden representations (Belinkov, 2022). Due to the many limitations of classifier or structural probe based probing techniques (Hewitt & Liang, 2019; White et al., 2021; Maudslay et al., 2020), we use Direct-Probe (Zhou & Srikumar, 2021), a non-classifier-based probing technique. DirectProbe clusters the hidden representations of a specific layer based on labels for the property we want to study. Then, the convex hull of these clusters (Figure 1f) can be used to study how well hidden representations encode information about that property. The basic idea is that a good-quality representation will have well-separated clusters, while linear encoding of a property will result in each label having one cluster. The quality of clustering can be evaluated by predicting clusters for a hold-out test set.

We provide details on Abstract Syntax Tree (AST), data flow graphs (DFG), motif structure and transformer model in Appendix B and transformer architecture in Appendix C.

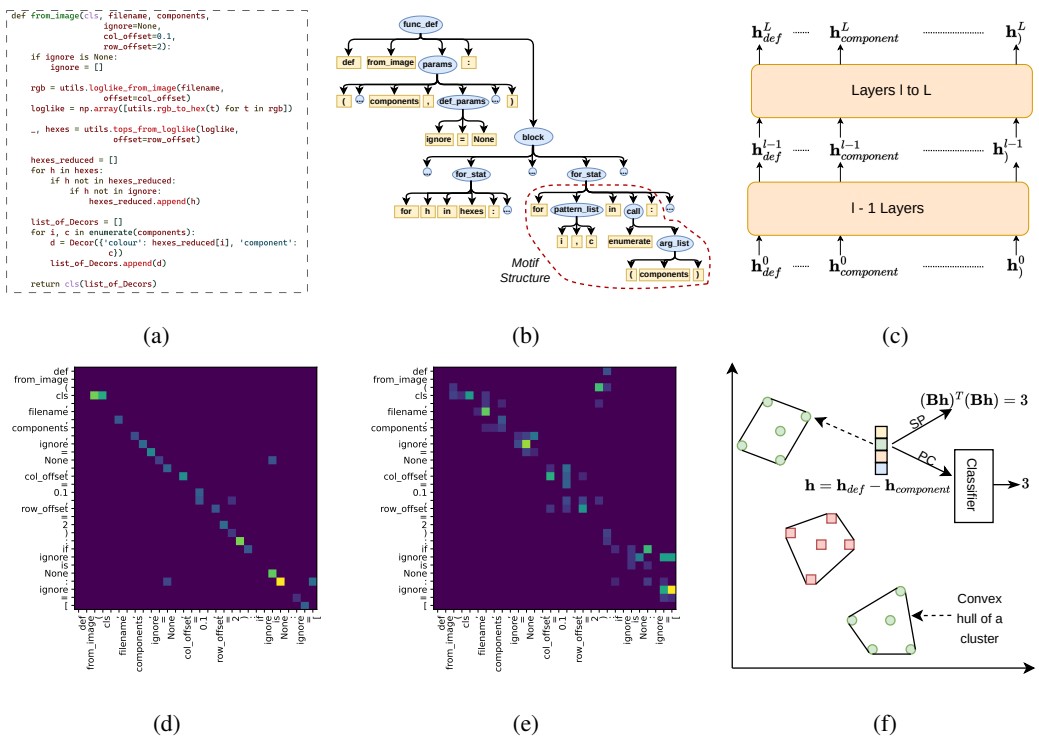

(a)                    (b)                    (c)

(d)                    (e)                    (f)

Figure 1: A python code snippet **(a)** and it's (partial) AST **(b)**; Illustration of hidden representation in a transformer model **(c)**; Attention map for head with best precision (head 1) **(d)** and head with best f-score (head 2) **(e)** of layer 9 of CodeBERT for first 30 tokens of the code; An illustration of structural probe (SP), probing classifier (PC) and convex hull created by DirectProbe **(f)**.

## 3 EXPERIMENTS

In this section, we elaborate on the experiments that we performed to analyze self-attention and hidden representation of PCMs. In attention analysis, we compare the self-attention of models with the relations in program's syntax tree and data flow graphs. For hidden representations, we perform probing without classifiers using DirectProbe (Zhou & Srikumar, 2021).

### 3.1 MODELS AND DATASET

We conducted experiments over five models: CodeBERT (Feng et al., 2020), GraphCodeBERT (Guo et al., 2021), UniXcoder (Guo et al., 2022), CodeT5 (Wang et al., 2021) and PLBART (Ahmad et al., 2021). PLBART consists of six encoder layers and was trained on Java and Python source codes obtained from Google BigQuery[1]. All other models comprise twelve encoder layers and were trained on the CodeSearchNet (CSN) dataset (Husain et al., 2019). UniXcoder's training uses flattened ASTs, while GraphCodeBERT incorporates Data Flow Graphs (DFGs) as part of its input. For our experiments, we randomly sampled 3000 Python codes from the test set of CSN dataset after removing docstrings and comments. More details about the models are presented in the Appendix D and about the dataset in the Appendix E.

### 3.2 ATTENTION ANALYSIS

#### 3.2.1 SETUP

**Model graph.** The attention map of a head is a $n * n$ matrix ($n$ is the number of input tokens). The elements of the matrix represent the significance each token attributes to other tokens. We consider

---

the matrix as the adjacency matrix of a graph with input tokens corresponding to nodes and attention values inducing an edge. Similar to previous works on attention analysis (Wan et al., 2022; Zhang et al., 2022a), we merge the sub-tokens of input code tokens by averaging their attention values.

Prior studies have typically set an arbitrary threshold of 0.3 for attention analysis and exclude heads with very few attention values, usually less than 100, from the analysis (Wan et al., 2022; Vig et al., 2021). This approach excludes more than 99.5% of self-attention values (see Appendix F), thereby skewing the conclusions drawn. For instance, Wan et al. (2022) reported high precision values, indicating that the majority of attention values correspond to relations in the AST. However, we observe a significantly reduced recall, as shown in Figure 2. Furthermore, the head with the highest precision often has next-token attention (Figure 1d), while the head with highest F-score encode complex relations (Figure 1e). So, to balance between precision and recall, we rely on F-score. We evaluate F-scores for all heads across various models and layers at different threshold values. Our analysis reveals that the highest F-score is achieved when using a threshold of 0.05 (as shown in Figure 3). We use this threshold for all experiments. Similar to previous works (Wan et al., 2022), all values below the threshold are set to 0 and above it to 1, i.e., edges are not weighted. Weighing the calculations with actual self-attention values will lower the precision and recall and increase the graph edit distance per node (Section 3.2.2). Setting values to 1 refers to the best-case scenario. Thus, the limitations documented in this work exists even in best-case scenario. Weighing with original values will only make these limitations more stark but the conclusion remains the same.

**Code graphs.** We create a syntax graph representing relations in an AST. Following Wan et al. (2022), we assume two tokens to have a syntactic relation if they exist in the same motif structure (Figure 1b). In DFG, we do not differentiate between the `ComesFrom` and `ComputedFrom` edges in attention analysis. We use the term code graphs to mean "both syntax and dataflow graphs". The syntax graph comprises syntactic relations among all tokens, while the DFG comprises relations among identifiers. Since we want to study encoding of syntactic-syntactic, identifier-identifier and syntactic-identifier relations separately, we create a non-identifier graph with the same nodes as syntax graph but only encompassing AST relations between syntactic tokens.

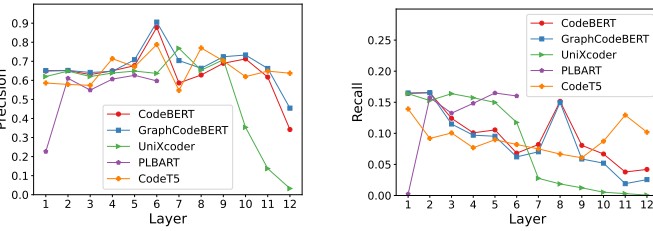

Figure 2: On comparing model graph with syntax graph with an attention threshold of 0.3, the precision (left) is high but the recall is very low (right).

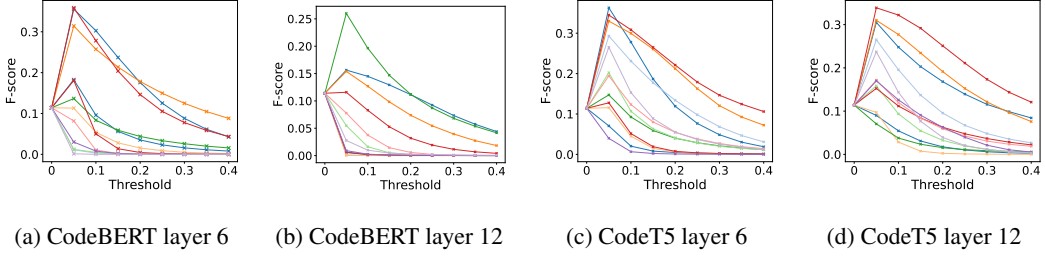

(a) CodeBERT layer 6     (b) CodeBERT layer 12     (c) CodeT5 layer 6     (d) CodeT5 layer 12

Figure 3: F-score between model graph and syntax graph at different thresholds for all heads. Each curve in a plot represents one head. For most heads, F-score is highest at a threshold of 0.05.

### 3.2.2 ANALYSIS

For each model, we compare the model graph of a head with each code graph in two ways. First, we compute the precision and recall between the set of edges in the model graph and code graphs. We

consider the edges of the code graphs as ground truth and those of the model graphs as predictions. For comparison across layers of a model, we select the heads with the highest F-score value for each layer. Second, we calculate graph edit distance (GED) (Sanfeliu & Fu, 1983) per node to quantify the similarity between the code graphs and model graphs. GED computes the cost of inserting, deleting, or substituting nodes and edges to transform one graph into an isomorphic graph of another. Code graphs and model graphs share the same set of nodes and have only one edge type. So, we assign a cost of 1 for both edge deletion and insertion operations and 0 otherwise. In all calculations, we apply the operations to model graphs. We also calculate the GED between the model graph and the non-identifier graph. For GED calculations, we use the NetworkX Python package (Hagberg et al., 2008).

## 3.3 Analysis of Hidden Representations

### 3.3.1 Qualitative Analysis with t-SNE

The hidden representation, $\boldsymbol{h}_i^l$ of $i^{th}$ word at the output of layer $l$, is a $d$-dimensional vector. We use t-distributed Stochastic Neighbor Embedding (t-SNE) (van der Maaten & Hinton, 2008) – a widely used technique to project high-dimensional data into a two-dimensional space while preserving the distance distribution between points - to qualitatively analyze the hidden representations of PCMs in two settings. First, we study the distribution of hidden representation of different token types; to this end, we collect the hidden representations of code tokens of specific token types from 100 programs, each having a minimum of 100 tokens. Second, we compare the distance distribution between tokens in an AST and between their hidden representations. To this end, we construct distance matrices of both for randomly selected code samples. We show their visualizations in Appendix G.

In the first setting, we find that the hidden representations form clusters based on token types rather than on syntactic relations. Moreover, as shown in Figure 5, the clusters of syntactically related tokens such as, `def`, `(`, `)` and `:`, do not exist close to each other. In the second setting, we first study the distance distribution between tokens in an AST. In the AST, siblings have similar distance distribution. So, in t-SNE visualization, siblings cluster together. If the distance between hidden representations corresponds to the distance in the AST, hidden representations should also have similar distance distribution. However, we observe that similar to the distribution of the hidden representations, the distance distributions also cluster by token types.

But in the case of distance matrix certain syntactically related tokens exist together. For the code in Figure 1a, we find that `def` is close to `(`, `)` and `:` and `if` is close to `is` and `none` in the visualization of fifth layer of CodeBERT (Figure 6). Similarly, `not` and `in` occur together. Identifiers, though, are far from syntactical tokens including the token `=`, which usually establishes relations among variables. We found similar patterns for other codes for deeper layers of all models, while in the first few layers, all tokens cluster together. We use DirectProbe (Zhou & Srikumar, 2021) to further investigate whether the hidden representations encode syntax and dataflow relations.

### 3.3.2 Probing on Hidden Representations

To experiment with DirectProbe, we create datasets for each layer of the models we examined. Each data point is represented as $(\boldsymbol{h}_i^l * \boldsymbol{h}_j^l) : label_t$. $* \in \{concatenation, difference\}$ is an operation between hidden representations of tokens $i$ and $j$ of layer $l$. $t \in \{siblings, treedistance, dataflow\}$ is a task to evaluate whether hidden representations encode the specific property. Each dataset is split in a $80 : 20$ ratio into training and test set. The training set is used to create clusters for each label and the test set is used to evaluate the quality of clustering. Moreover, we look at the number of clusters to study if the hidden representations encode relations linearly or non-linearly.

Using data flow, we study hidden representations with respect to data flow relations. Here, both $i$ and $j$ are identifiers, $label \in \{NoEdge, ComesFrom, ComputedFrom\}$ and $* = concatenation$. Using siblings and tree distance, we study encoding of relations in an AST. For both tasks, token $i$ is one of a subset of Python keywords (detailed in Appendix H). In one set of experiments (`Keyword-All`), token $j$ can be any other token. In another set (`Keyword-Identifier`), token $j$ is an identifier. For siblings task, $label \in \{sibling, notsibling\}$, where two tokens in the same motif structure are considered to be siblings, and $* = concatenation$.

The tree distance between a keyword and an identifier denote different identifier types and syntax structure. For example, for the code in Figure 1b, the identifier types function name, parameters and default parameters are at a distance of 2, 3 and 4 from `def` respectively. Similarly, `for` is at a distance of 2 from the variable `h` when the iterator is also a variable (`hexes`) but at a distance of 3 from variables `i` and `c` when iterator has a function call, `enumerate`. Thus, distance between tokens varies with different syntax. Hence, if the hidden representations encode information about different identifier types and syntax, it follows that hidden representations of `Keyword-Identifier` pairs at a certain distance in AST must form separable clusters. The minimum distance between two code tokens in an AST is 2 while, tokens far apart in an AST don't have any discriminative syntactic relations. So, for tree distance, we only consider $label \in \{2, 3, 4, 5, 6\}$. Moreover, Reif et al. (2019) showed that square of distance between two vectors, $(\boldsymbol{h}_i^l - \boldsymbol{h}_j^l)^T (\boldsymbol{h}_i^l - \boldsymbol{h}_j^l)$, corresponds to distance in a tree. Hence, we set $* = difference$ for distance prediction task.

## 4 RESULTS AND DISCUSSION

### 4.1 WHAT DO MODELS ENCODE?

We present the results of the attention analysis in Figure 4 and those of DirectProbe in Tables 1, 2 and 3 for last layer. Plots in Figures 4a and 4b show that PCMs encode both syntactic and data flow relations within self-attention values. We also find that the middle layers encode these relations better than the deeper layers. This contradicts prior studies Wan et al. (2022), which concluded that the last two layers encode the relations in an AST better. The different conclusion results from using a lower threshold and from comparing the heads with the highest F-score, instead of comparing precision across layers of a model. Furthermore, in Figure 4c, we observe that model graphs at each layer exhibit a high degree of similarity with DFG, requiring less than one insertion or deletion per node in the model graphs. When considering the later layers of CodeBERT, UniXcoder, and PLBART, we observe an even higher level of similarity with the DFG. This contrasts with the low recall values in Figure 4a. A low recall value indicates the need for more insertions, hence higher insertion cost. Since the total cost is low, the deletion cost for later layers must be very low. Therefore, later layers of these models have fewer self-attention edges and correspond mostly to data-flow relations. That is, later layers retain fewer but more specialized code relations.

Table 9 shows that DirectProbe can form separable clusters for siblings with high label accuracy, except for UniXcoder. Also, for data-flow edge prediction, we observe separable clusters (Table 10). In this case, however, very poor accuracy for `NoEdge` shows that hidden representations can understand when an edge exists, including the edge type, but not when an edge doesn't exist. This could be because PCMs encode relations other than data-flow too.

### 4.2 HIDDEN REPRESENTATION ENCODE INFORMATION NON-LINEARLY

The number of clusters created by DirectProbe indicates whether the hidden representations encode a property linearly or non-linearly. Specifically, linear encoding results in the same number of clusters as the number of labels. For all three tasks, we observe a significantly higher number of clusters than labels, usually twice as many, as shown in Tables 1, 2, and 3 (also in Appendix I for layers 5 and 9). Thus, hidden representations encode syntactic and data flow relations non-linearly. Moreover, the distance between the convex hulls of the clusters is very low for siblings and data flow tasks and zero for the tree distance prediction task. Thus, finding a decision boundary is not trivial. This implies that probing requires a complex probe - a complex classifier (Zhou & Srikumar, 2021) or a non-linear structural probe (White et al., 2021) - to study hidden representations of PCMs.

However, previous studies that utilize probes (Karmakar & Robbes, 2021; Troshin & Chirkova, 2022; Wan et al., 2022; López et al., 2022) to study hidden representations, use a simple probe. Yet, they show that their probe is sufficient to investigate hidden representations. The conclusion of previous works could be incorrect due to the probe itself learning the task, similar to probes learning linguistic tasks in NLP (Hewitt & Liang, 2019), rendering the result of probing meaningless. Hewitt & Liang (2019) suggested the use of control tasks to ensure that the probe is selective with high accuracy on linguistic tasks and low accuracy on control tasks. Hence, control tasks must accompany the use of probes, when investigating hidden representations.

### 4.3 LIMITATIONS OF PCMs

The limitations of current PCMs become apparent when we study the syntactic-syntactic, identifier-identifier, and syntactic-identifier relations separately. We use the similarity analysis to study each kind of relation separately (see Figure 4c). We observe that for different layers, the non-identifier graph has a similarity score per node of approximately 1.5 for CodeBERT, GraphCodeBERT, and UniXcoder, between 2.0 to 2.5 for PLBART, and between 2.0 to 3.0 for CodeT5. The set of edges in a non-identifier graph is a subset of edges in the syntax graph. If the additional edges in the syntax graph that are not in the non-identifier graph were present in the model graphs, the deletion cost and hence the overall cost for the syntax graph would have decreased. However, we observe a significant increase in cost per node, by a factor of 1.5 - 2 times. The additional edges under consideration relate syntactic and identifier tokens. The fact that they seem not to be present in the model graph indicates that they are not encoded in the self-attention values.

For `Keyword-Identifier` token pairs, we observe very poor clustering accuracy for $distance > 2$ (see Table 8). Thus the hidden representations do not encode sufficient information for the distance prediction task. As described in Section 3.3.2, this implies that hidden representations of PCMs do not encode information about different identifier types and syntax structures. Thus, self-attention values do not encode relations between syntactic and identifier tokens, resulting in hidden representations being unable to discriminate between different identifier types and syntax structures, e.g., between a function name and a parameter in relation to `def`, or between a variable and a function call in relation to `for` or `if`. The first example limits PCMs in understanding subtle syntactic differences; the second limits both, syntactic and logical understanding of code.

Despite these limitations, PCMs perform well on benchmarks. However, on real-world tasks or with slight distribution shift from training data the models fail to generalize (Rabin et al., 2023a; Yang et al., 2023b). Allal et al. (2023) showed that simple filtering of training data by using repositories with 5+ GitHub stars, commonly used as a proxy for code quality, is sufficient to deteriorate performance on code generation and fill-in-the-middle tasks due to distribution shift between real-world usage and high-quality code. Other works have documented that PCMs depend on shortcuts such as function names, comments and variables for predictions, such as generating code summaries, instead of exploiting program logic (Sontakke et al., 2022). Shortcut learning (Du et al., 2021; Geirhos et al., 2020) to achieve good performance on benchmarks has been well documented in deep learning.

As discussed above, PCMs fail to encode sufficient information to understand subtle syntactic differences and code logic. Without proper comprehension of code, PCMs have to rely on shortcut cues for prediction. Thus, shortcuts in PCMs arise due to their inability to encode relevant code information. Since these shortcuts may not be available outside of training data, PCMs fail to generalize. For instance, models can fail when developers do not use proper function names, they do not comment their code properly, or comments and function names of real code differ from the patterns seen by the model during training.

### 4.4 THE EFFECT OF INPUT AND PRE-TRAINING OBJECTIVES

While all PCMs encode syntactic and dataflow relations, they do it to varying degrees. Among the considered models, GraphCodeBERT, UniXcoder and CodeT5 include AST or DFG information either in their input, in the pre-training objectives, or in both. GraphCodeBERT is trained with DFG as part of its inputs along with pre-training objectives to learn the representation from the data flow, CodeT5 is trained with identifier-aware pre-training objectives (Wang et al., 2021) to learn information about identifiers, while UniXcoder is trained with flattened AST as part of its input.

Both GraphCodeBERT and CodeT5 exhibit high recall with DFG, even for the last two layers. Thus they encode a high proportion of DFG edges in the self-attention of final layers. However, CodeT5 has very low similarity with DFG (high cost per node), while GraphCodeBERT has high similarity (Figure 4c). The higher cost can only be explained by significantly higher deletion cost for CodeT5. Thus, the two training conditions make the PCMs encode information about DFG in the later layers, but identifier-aware pre-training leads to encoding non-DFG specific relations, too.

Surprisingly, among all models, the later layers of UniXcoder encode the least information about syntactic relations in both the self-attention values and in hidden representations. Moreover as shown in Table 9, it is the only model whose hidden representation of the final layer does not perform well

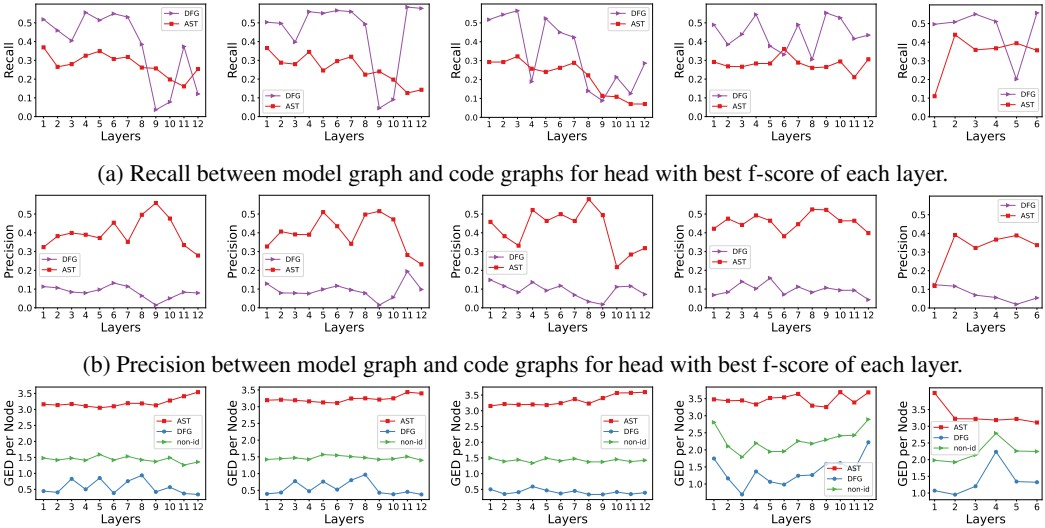

(a) Recall between model graph and code graphs for head with best f-score of each layer.

(b) Precision between model graph and code graphs for head with best f-score of each layer.

(c) Graph edit distance (GED) per node of model graph from different code graphs for each layer.

Figure 4: Results of attention analysis for all encoder layers of (from left to right) CodeBERT, GraphCodeBERT, UniXcoder, CodeT5 and PLBART.

on the siblings prediction task. Thus, the use of flattened ASTs negatively impacts the ability of a model to learn code syntax.

Based on these observations and the comparison between the training of GraphCodeBERT and CodeT5 with the training of UniXcoder, we suggest the use of syntax-aware pre-training objectives to help models encode more syntactic information. In Section 4.3, we also observed that PCMs fail to discriminate between different identifier types which hinders their ability to understand differences in code syntax and code logic. Training models to predict different identifier types will force them to encode this information. Encoding subtleties of code and making predictions based on these will limit the dependence of PCMs on shortcuts and help them generalize better to real-world tasks.

Table 1: Analysis of hidden representation of last layer on AST distance prediction with 5 labels. We report the number of clusters formed by DirectProbe, distance between clusters, and label accuracy.

| Tokens | Model | No. of clusters | Distance | | Label Accuracy | | | | |
|--------|-------|-----------------|----------|----------|----------------|------|------|------|------|
| | | | Min | Avg | 2 | 3 | 4 | 5 | 6 |
| {Keyword-All} | CodeBERT | 10 | 0.0 | 1.27 | 0.85 | 0.75 | 0.73 | 0.68 | 0.55 |
| | GraphCodeBERT | 9 | 0.0 | 1.30 | 0.84 | 0.78 | 0.67 | 0.67 | 0.57 |
| | UniXcoder | 13 | 0.0 | 2.60 | 0.41 | 0.55 | 0.42 | 0.48 | 0.51 |
| | CodeT5 | 10 | 0.0 | 1.38 | 0.83 | 0.79 | 0.70 | 0.64 | 0.60 |
| | PLBART | 9 | 0.0 | 1.88 | 0.83 | 0.83 | 0.77 | 0.70 | 0.60 |
| {Keyword-Identifier} | CodeBERT | 7 | 0.0 | 0.53 | 0.82 | 0.66 | 0.56 | 0.53 | 0.51 |
| | GraphCodeBERT | 7 | 0.0 | 2.32 | 0.79 | 0.68 | 0.52 | 0.57 | 0.49 |
| | UniXcoder | 9 | 0.0 | 5.38 | 0.37 | 0.49 | 0.36 | 0.32 | 0.34 |
| | CodeT5 | 6 | 0.0 | 3.09 | 0.78 | 0.66 | 0.59 | 0.55 | 0.48 |
| | PLBART | 5 | 0.0 | 0.10 | 0.84 | 0.73 | 0.52 | 0.66 | 055 |

# 5    RELATED WORK

**Explainability.** Several studies have tried to explain the working of PCMs. Cito et al. (2022) and Rabin et al. (2023b) used input perturbation, while, Liu et al. (2023) used backpropagation to find the most relevant input tokens. Zhang et al. (2022a) created an aggregated attention graph and studied its application to the `VarMiuse` task. Wan et al. (2022) performed attention analysis and probing with structural probes (Hewitt & Manning, 2019). López et al. (2022) used structural probe to create binarized AST from hidden representations. Probing classifiers have been used to test syntax and semantic understanding (Karmakar & Robbes, 2021; Troshin & Chirkova, 2022; Ahmed et al., 2023), the effect of positional embeddings (Yang et al., 2023a), the relation between self-attention

Table 2: Analysis of hidden representation of last layer on siblings prediction with 2 labels.

| Tokens | Model | No. of clusters | Distance | | Label Accuracy | |
|---|---|---|---|---|---|---|
| | | | Min | Avg | Not Siblings | Siblings |
| {Keyword-All} | CodeBERT | 3 | 0.14 | 0.45 | 0.87 | 0.88 |
| | GraphCodeBERT | 4 | 0.06 | 6.95 | 0.76 | 0.87 |
| | UniXcoder | 4 | 0.05 | 28.73 | 0.61 | 0.64 |
| | CodeT5 | 7 | 0.0 | 1.80 | 0.82 | 0.91 |
| | PLBART | 5 | 0.58 | 4.89 | 0.88 | 0.88 |
| {Keyword-Identifier} | CodeBERT | 4 | 0.18 | 4.62 | 0.79 | 0.87 |
| | GraphCodeBERT | 3 | 0.14 | 0.54 | 0.75 | 0.86 |
| | UniXcoder | 3 | 0.0 | 3.13 | 0.47 | 0.56 |
| | CodeT5 | 4 | 0.0 | 0.42 | 0.80 | 0.86 |
| | PLBART | 4 | 0.28 | 5.17 | 0.80 | 0.87 |

Table 3: Analysis of hidden representation of last layer on data flow edge prediction with 3 labels.

| Tokens | Model | No. of clusters | Distance | | Label Accuracy | | |
|---|---|---|---|---|---|---|---|
| | | | Min | Avg | No Edge | ComesFrom | ComputedFrom |
| {Identifier-Identifier} | CodeBERT | 4 | 0.24 | 3.68 | 0.69 | 0.91 | 0.90 |
| | GraphCodeBERT | 7 | 0.0 | 8.61 | 0.71 | 0.94 | 0.93 |
| | UniXcoder | 4 | 0.92 | 12.71 | 0.57 | 0.72 | 0.79 |
| | CodeT5 | 4 | 0.29 | 3.09 | 0.57 | 0.86 | 0.90 |
| | PLBART | 4 | 0.72 | 8.99 | 0.62 | 0.91 | 0.94 |

and distance in AST (Chen et al., 2022) and logic understanding (Baltaji & Thakkar, 2023). Other studies have established correlations between input tokens, model output, and self-attention. Bui et al. (2019) created an attention-based discriminative score to rank input tokens and studied the impact of high-ranked tokens on output. Attention-based token selection was utilized Zhang et al. (2022b) to simplify the input program of CodeBERT (Feng et al., 2020). Rabin et al. (2021) and Rabin et al. (2022) simplified the input program while preserving the output and showed that the percentage of common tokens between attention and reduced input programs is typically high.

**Limitations of PCMs.** Hellendoorn et al. (2019) and Aye et al. (2021) reported a substantial performance gap when PCMs are applied to real-world tasks. Hajipour et al. (2022) showed that out-of-distribution scenarios are challenging for most PCMs. Sontakke et al. (2022) observed that PCMs rely heavily on comments, function names, and variable names for code summarization; masking subtle code logic does not change the generated summaries. Rabin et al. (2023a) showed that models can memorize training data, and Yang et al. (2023b) demonstrated that the memorization increases with model size. Moreover, Barone et al. (2023) observed that all models they analyzed struggled with "statistically uncommon correct continuations".

**Our Work** also studies limitations of PCMs. However, unlike previous work on this topic, which only document the limitations, we provide explanations as to why these limitations might exist. Regarding explainability, we critically examine arbitrary assumptions made in previous works and show that they can result in misleading or wrong conclusions.

## 6 CONCLUSION

We presented a comprehensive analysis of the self-attention mechanism and the hidden representations within PCMs with respect to their role in enabling PCMs to capture structural and data flow relations in code. Additionally, we critically studied arbitrary assumptions made in previous work. Our findings shed light on a significant limitation of PCMs – the attention maps do not encode syntactic-identifier relations between code tokens. Furthermore, we show that this limitation results in the inability of hidden representations to distinguish between different identifier types and syntax structures. We believe that this limitation is what leads to syntactic errors in outputs of PCMs. Moreover, this limitation also causes PCMs to not generalize to scenarios where information about syntactic-identifier relations are necessary. Insights form our work will lead to more robust experimental designs for model interpretability and improvement in training methods, such as syntax-aware pre-training objective, to mitigate the limitations that we revealed. In future work, we intend to expand upon the presented study to consider larger models and to explore the NL-PL alignment. This extension will enable us to investigate more recent instruction-tuned models.

## REPRODUCIBILITY STATEMENT

The models and dataset used in our work are open-sourced and available with permissible licenses. We provide the details of models in Appendix D and dataset pre-processing details in Appendix E. We present the steps to create the dataset for experiments with DirectProbe in Appendix H. The experimental setting for t-SNE is presented in Appendix G. Finally, we share the code, as well as, the results as supplementary material.

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

## A  HARDWARE DETAILS

We first perform a forward pass through the models on an Nvidia A6000 48GB GPU and store the attention and hidden representation for experiments. All experiments are then run on an AMD Ryzen Threadripper 5975WX with 32 cores.

## B  ADDITIONAL BACKGROUND DETAILS

**Abstract Syntax Trees (ASTs)** are data structures that represent the syntactic structure of a code. The leaf nodes of the tree represent code tokens, and internal nodes represent different constructs of the code such as `if-else` block, `identifiers`, or `parameters`. A partial AST[2] for a Python code snippet is shown in Figure 1b.

**Data Flow Graphs (DFGs)** have nodes representing variables and edges depicting how the values flow from one variable to another. We adopt the approach by Guo et al. (2021) to obtain the data flow relations, with two types of data flow relations, viz. `ComesFrom` and `ComputedFrom`.

**Motif Structure** Wan et al. (2022) defines motif structure as a non-leaf node in the AST with all it's children. We show motif structure in Figure 1b.

**Transformer and Self-attention.** A Transformer model consists of $L$ stacked transformer blocks. The core mechanism of a transformer block is self-attention. Given a code $c = \{c_1, c_2, ..., c_n\}$ of length $n$, the self-attention mechanism assigns an input token $c_i$ attention values over all input tokens. The code $c$ is first transformed into a list of $d$-dimensional vectors $\boldsymbol{H}^0 = [\boldsymbol{h}_1^0, \boldsymbol{h}_2^0, ..., \boldsymbol{h}_n^0]$. The transformer model transforms $\boldsymbol{H}^0$ into a new list of vectors $\boldsymbol{H}^L$. A layer $l$ takes the output of the previous layer $\boldsymbol{H}^{l-1}$ as input and computes $\boldsymbol{H}^l = [\boldsymbol{h}_1^l, \boldsymbol{h}_2^l, ..., \boldsymbol{h}_n^l]$. $\boldsymbol{h}_i^l$ is the hidden representation of $i^{th}$ word at layer $l$, as shown in Figure 1c. Attention values for layer $l$ are computed as

$$Attention(\boldsymbol{Q}, \boldsymbol{K}, \boldsymbol{V}) = softmax(\frac{\boldsymbol{Q}\boldsymbol{K}^T}{\sqrt{d}})\boldsymbol{V} \tag{1}$$

where $\boldsymbol{Q} = \boldsymbol{H}^{l-1}\boldsymbol{W}_Q^l$, $\boldsymbol{K} = \boldsymbol{H}^{l-1}\boldsymbol{W}_K^l$ and $\boldsymbol{V} = \boldsymbol{H}^{l-1}\boldsymbol{W}_V^l$. In practice, a layer $l$ contains multiple heads, each with its own $\boldsymbol{W}_Q^l$, $\boldsymbol{W}_K^l$, $\boldsymbol{W}_V^l$ matrices. Each head thus has a set of attention values among each pair of input tokens, which constitute the attention map for that head (Figure 1e).

---

[2]We use tree-sitter (https://tree-sitter.github.io/tree-sitter/)to obtain AST of a code.

## C  TRANSFORMER ARCHITECTURE

A Transformer model has encoder and decoder blocks and can be an encoder-only, a decoder-only, or an encoder-decoder model (Xu & Zhu, 2022). The encoder builds a continuous representation of inputs while the decoder generates a target sequence. So, the encoder is optimized for understanding input while the decoder is optimized for generation. Building a good low-dimensional representation of input data is important for efficient learning. This is missing from a decoder-only model. While the decoder-only model works well despite this limitation, the gains in performance are only observed at a very high scale. Well-performing decoder-only models have more than a billion parameters.

However, the performance can be easily matched by encoder-only or encoder-decoder models with significantly fewer parameters. For example, CodeT5+ (Wang et al., 2023) with 220M parameters matches the much larger CodeGen-mono (Nijkamp et al., 2023) with 2B parameters on the GSM8K dataset (Cobbe et al., 2021). Further, the application of decoder-only models remains limited to generation tasks. For instance, consider Li et al. (2023), which contributed a decoder-only model for code generation. The authors remove personally identifiable information (PIIs) from the dataset before training. To remove PIIs, the authors train a 125M encoder-only model, named StarEncoder, and not a decoder-only model.

Decoder-only models are limited to code generation due to uni-directional / non-causal training. Training with fill-in-the-middle (Bavarian et al., 2022) alleviates this issue somewhat and improves performance but the training still remains auto-regressive which prevents the inclusion of code properties such as AST and data flow relations in training inputs and objectives. In contrast, encoder-only and encoder-decoder models have significant variations in training strategies with different inputs and pre-training objectives. Models such as GraphCodeBERT (Guo et al., 2021) and UniXcoder (Guo et al., 2022) are also trained on data flow graphs and AST respectively.

We wanted to understand what a PCM does and does not understand and how the understanding is affected by different pre-training strategies. Also, we believe that smaller encoder-only and encoder-decoder models can be better trained to improve their performance and generalizations since more information can be incorporated into their training objectives. So, we selected five relatively small encoder-only and encoder-decoder models. Since the encoder is optimized to build a low-dimensional representation of input data, we perform analysis over encoder attentions.

These models were carefully selected to encompass different pre-training objectives. The details of these models are in Appendix D.

Finally, our analysis of GraphCodeBERT and CodeT5 shows that including code property-based objectives helps the model learn code properties better. However, including such objectives with decoder-only model is yet to be explored. So, we did not extend the analysis to decoder-only models.

One practical limitation of smaller models is their inability to learn in-context. However, Xie et al. (2022) created a synthetic dataset and showed that LSTM and Transformer models with less than 200M parameters can learn in-context after training on the synthetic dataset. Thus, it is possible to make smaller models learn in-context too. Combining improved training strategies suggested by our work with training on such synthetic dataset has the potential to train smaller models with in-context learning ability and better generalizability.

## D  MODEL DETAILS

We ran our experiments with 5 models - CodeBERT (Feng et al., 2020), GraphCodeBERT (Guo et al., 2021), UniXcoder (Guo et al., 2022), CodeT5 (Wang et al., 2021) and PLBART (Ahmad et al., 2021).

**CodeBERT** is an encoder-only bi-directional transformer with 12 layers, each layer having 12 heads. It has been trained on CodeSearchNet (CSN) (Husain et al., 2019) dataset with two pretrained objectives. Masked Language Modeling (MLM) objective is used with bimodal (NL-PL pair) data, the model is trained with and Replaced Token Detection (RTD) with unimodal (only PL) data.

**GraphCodeBERT** uses the same architecture as CodeBERT but also takes nodes of the data flow graph (DFG) of the code as inputs with special position embeddings to indicate which tokens are nodes of DFG. It is also trained on CSN dataset. The model is first trained with MLM objective, followed by edge prediction in data flow graph and node alignment between code tokens and DFG nodes.

**UniXcoder** is an encoder-decoder model. However, the model can be used in encoder-only, decoder-only or encoder-decoder mode using a special input token, [MODE]. It is also trained on CSN dataset and taked flattened ASTs of code as part of it's input during training. The model is trained with masked spans prediction, masked language modeling, multi-modal contrastive learning, whereby positive pairs are created using dropout, and cross-modal generation.

**CodeT5** is an encoder-decoder model trained on CSN dataset with identifier-aware and bimodal-dual generation objective. Identifier-aware pretraining uses masked span prediction, identifier tagging and masked identifier prediction alternatively to make the model attend to identifiers while bimodal-dual generation consists of NL to PL generation and PL to NL generation.

**PLBART** PLBART is an encoder-decoder model with 6 encoder layers, each with 12 heads. The model is trained with 3 denoising objectives - token masking, token deletion and token infilling - on NL and PL data from Google BigQuery[3].

For details on various pre-training objectives, refer to Xu & Zhu (2022).

## E  DATSET DETAILS

Since four of the five models we experimented with were trained on CSN dataset, we chose the python codes from test split of CSN for our experiments. CSN dataset consists of 2 million comment-code pairs from 6 programming languages. The programming languages are Go, Java, JavaScript, PHP, Python and Ruby. The models trained on CSN are pre-trained with unimodal and bimodal objectives on multiple programming languages.

Before performing analysis we pre-process the dataset by removing any doctring and code comments from the dataset. CodeBERT, GraphCodeBERT and UniXcoder has a maximum input token length of 512 tokens. So, we create a subset consisting of codes with less than 500 tokens post tokenization. CSN consists a list of code tokens for each token. For merging attention and hidden representation of sub-tokens, we use this list to keep track of where a token has been split by tokenizer. However, the list splits `*args` into `*` and `args` and `**kwargs` into `*`, `*` and `kwargs`. in python, `*` is used for iterator unpacking and `**` for dictionary unpacking. So, to differentiate the two, we merger the `*`s of `kwargs`. Then, we randomly sample 3000 python code and run our experiments on these codes.

## F  ATTENTION DISTRIBUTION

In Table 4, we present the percentage of attention values which are 0, between 0 - 0.05, between 0.05 - 0.3 and more than 0.3. Note that we assume any value below 0.001 to be 0.

Table 4: Percentage of attention values in differenr range.

| Range | CodeBERT | GraphCodeBERT | UniXcoder | CodeT5 | PLBART |
|-------|----------|---------------|-----------|--------|--------|
| 0.0 | 59.13 | 70.3 | 67.28 | 51.92 | 74.63 |
| 0.0 - 0.05 | 39.25 | 28.58 | 31.88 | 46.23 | 74.27 |
| 0.05 - 0.3 | 1.48 | 1.00 | 0.76 | 1.64 | 0.97 |
| above 0.3 | 0.14 | 0.12 | 0.08 | 0.22 | 0.13 |

---

[3]https://console.cloud.google.com/marketplace/details/github/github-repos

## G   T-SNE

We select 100 codes with at least 100 code tokens and get the hidden representation for each token. We then select hidden representation of the token types shown in Figure 5. We ran t-SNE on the selected hidden representation with different perplexity value (van der Maaten & Hinton, 2008) from 5 to 50 for all layers of all models. Increasing the perplexity value only made the clusters tighter but the overall distribution of points remained similar. So, the conclusion is not affected by perplexity value. We set the number of iterations to 50K, ensuring t-SNE always converges (no change in error for at least 300 iterations). We found that for all layers, tokens of same type were closer, though the clustering of same token types became tighter for deeper layers. We show the visualization for fifth layer of CodeBERT with perplexity of 50 in Figure 5.

We create a distance matrix for both the tree distance in AST and distance between hidden representation of tokens for a few code. We run t-SNE till convergence with perplexity values 5 and 10 and found the distribution to be similar. We again observed clusters of tokens of same types for hidden representation, unlike clusters of AST distance matrix. The clusters are closer for earlier layers and farther for deeper layers. We show the visualization for fifth layer of CodeBERT for code in Figure 1a in Figure 6.

We use the t-SNE implementation provided by the sci-kit learn library[4].

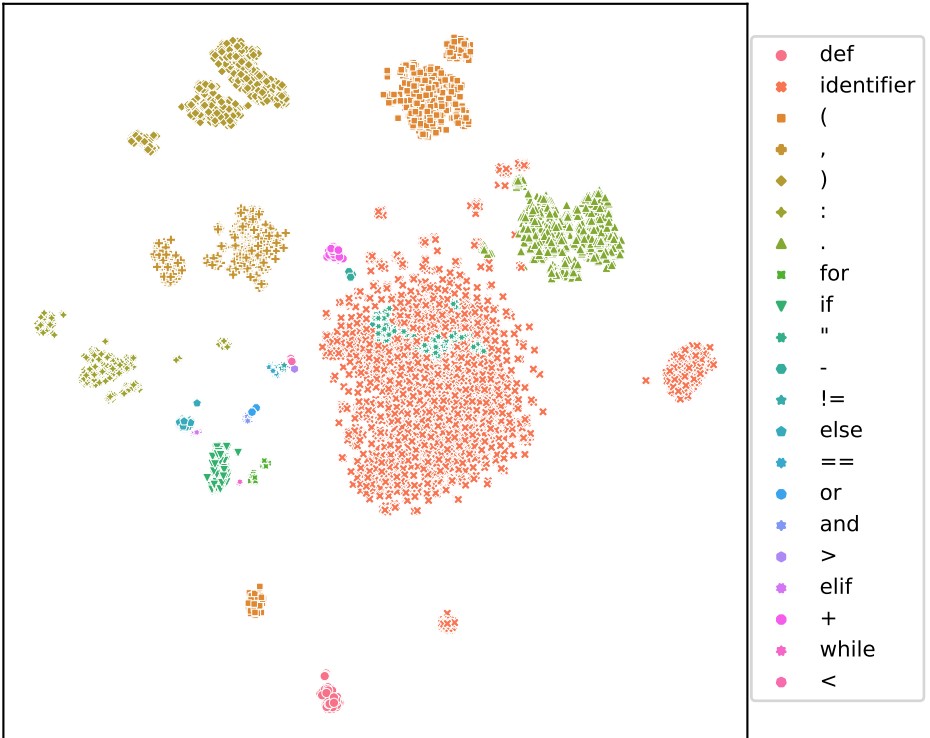

Figure 5: t-SNE visualization of hidden representation of layer 5 of CodeBERT for selected token types.

## H   DIRECTPROBE EXPERIMENT DETAILS

For siblings and tree distance prediction tasks, the first token is of one of the following to-ken types: `def for if none else false true or and return not elif with try raise except break while assert print continue class`.

---

[4]https://scikit-learn.org/stable/modules/generated/sklearn.manifold.TSNE.html

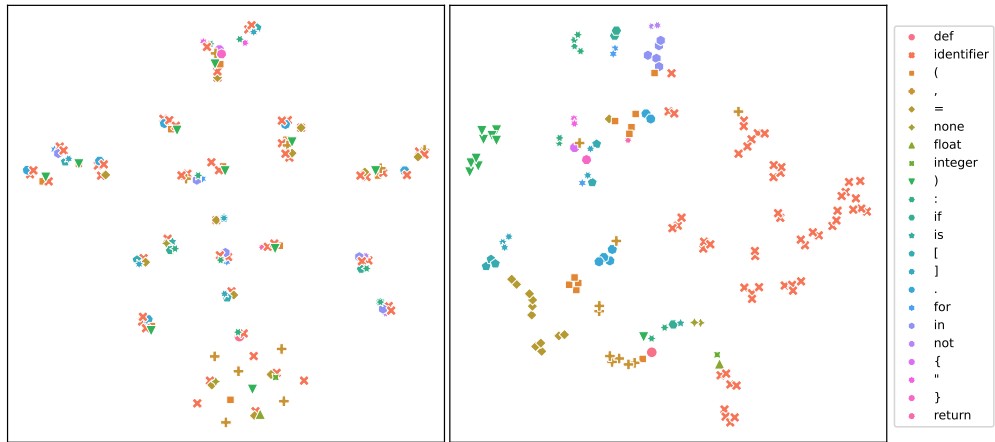

Figure 6: t-SNE visualization of distance matrix for AST(left) and hidden representation (right) of layer 5 of CodeBERT for code in Figure 1a.

For distance prediction task, we randomly sample 160 codes. We select the code pairs at a maximum distance of 6, ensuring first token is of one of the selected tokens types. The second token can be of any type. We then select 1300 code pairs for each layer resulting in a dataset of 6500 data points. We split it into train and test set in the ration of 80:20. We follow the same steps for `Keyword-Identifier` too, with the difference that we use 450 codes and the second token is of type `identifier`.

For distance prediction task, we randomly sample 100 codes. We first select all tokens which are one of the selected token types. We then select equal number of siblings and non-siblings for each of these selected tokens. From this, we randomly sample 1500 siblings and 1500 non-siblings resulting in 3000 data points. We split it into train and test set in the ration of 80:20. We follow the same steps for `Keyword-Identifier` too, with the difference that we use 300 codes and the second token is of type `identifier`.

For data flow edge prediction task, we randomly sample 130 codes. We first select an identifier and then the tokens which has a data flow edge with the first token. We then select $n$ tokens which do not have data flow edge with the first token, where, $n = max(num(ComesFrom), num(ComputedFrom))/2$. From the selected pairs, we randomly sample 1500 pairs for each label resulting in 4500 data points. We split it into train and test set in the ration of 80:20.

In all tasks, we ensure that the same data points are used for all models and layers.

## I  DIRECTPROBE RESULTS AND CLUSTER STATISTICS

In this section, we provide the statistics of size and label of cluster created by DirectProbe for last layer of each model for each and the results of experiments with DirectProbe for layer 3 of PLBART and layer 5 and layer 9 of other models. Analysis with DirectProbe is presented in Tables 5, 6 and 7. The cluster statistics are presented in Tables 8, 9 and 10.

Table 5: Results of analysis by DirectProbe for tree distance prediction with 5 labels.

| Tokens | Model (Layer) | No. of clusters | Distance | | Label Accuracy | | | | |
|---|---|---|---|---|---|---|---|---|---|
| | | | Min | Avg | 2 | 3 | 4 | 5 | 6 |
| {Keyword-All} | CodeBERT (5) | 9 | 0.0 | 1.09 | 0.87 | 0.85 | 0.74 | 0.72 | 0.62 |
| | CodeBERT (9) | 9 | 0.0 | 1.36 | 0.89 | 0.81 | 0.72 | 0.72 | 0.61 |
| | GraphCodeBERT (5) | 11 | 0.0 | 3.99 | 0.88 | 0.84 | 0.75 | 0.70 | 0.63 |
| | GraphCodeBERT (9) | 9 | 0.0 | 1.74 | 0.83 | 0.81 | 0.69 | 0.68 | 0.62 |
| | UniXcoder (5) | 10 | 0.0 | 1.87 | 0.86 | 0.82 | 0.72 | 0.71 | 0.66 |
| | UniXcoder (9) | 9 | 0.0 | 0.70 | 0.77 | 0.77 | 0.69 | 0.63 | 0.63 |
| | CodeT5 (5) | 9 | 0.0 | 1.65 | 0.79 | 0.80 | 0.70 | 0.67 | 0.65 |
| | CodeT5 (9) | 13 | 0.0 | 8.50 | 0.85 | 0.83 | 0.64 | 0.70 | 0.67 |
| | PLBART (3) | 13 | 0.0 | 2.60 | 0.79 | 0.77 | 0.62 | 0.70 | 0.57 |
| {Keyword-Identifier} | CodeBERT (5) | 5 | 0.0 | 0.06 | 0.86 | 0.74 | 0.64 | 0.68 | 0.59 |
| | CodeBERT (9) | 7 | 0.0 | 3.41 | 0.89 | 0.77 | 0.63 | 0.65 | 0.57 |
| | GraphCodeBERT (5) | 5 | 0.0 | 0.05 | 0.83 | 0.70 | 0.63 | 0.64 | 0.56 |
| | GraphCodeBERT (9) | 7 | 0.0 | 2.79 | 0.83 | 0.69 | 0.60 | 0.62 | 0.56 |
| | UniXcoder (5) | 7 | 0.0 | 2.33 | 0.82 | 0.66 | 0.61 | 0.61 | 0.49 |
| | UniXcoder (9) | 7 | 0.0 | 5.07 | 0.69 | 0.61 | 0.53 | 0.55 | 0.44 |
| | CodeT5 (5) | 7 | 0.0 | 2.42 | 0.68 | 0.59 | 0.53 | 0.54 | 0.45 |
| | CodeT5 (9) | 5 | 0.0 | 0.23 | 0.78 | 0.66 | 0.60 | 0.61 | 0.51 |
| | PLBART (3) | 9 | 0.0 | 7.48 | 0.66 | 0.59 | 0.49 | 0.49 | 0.46 |

Table 6: Results of analysis by DirectProbe for siblings prediction with 2 labels.

| Tokens | Model (Layer) | No. of clusters | Distance | | Label Accuracy | |
|---|---|---|---|---|---|---|
| | | | Min | Avg | Not Siblings | Siblings |
| {Keyword-All} | CodeBERT (5) | 4 | 0.19 | 8.75 | 0.87 | 0.94 |
| | CodeBERT (9) | 4 | 0.23 | 8.55 | 0.87 | 0.93 |
| | GraphCodeBERT (5) | 5 | 0.24 | 8.38 | 0.87 | 0.91 |
| | GraphCodeBERT (9) | 4 | 0.24 | 3.30 | 0.84 | 0.92 |
| | UniXcoder (5) | 4 | 0.20 | 9.62 | 0.86 | 0.91 |
| | UniXcoder (9) | 4 | 0.14 | 6.73 | 0.80 | 0.88 |
| | CodeT5 (5) | 5 | 0.17 | 17.09 | 0.84 | 0.85 |
| | CodeT5 (9) | 5 | 0.70 | 16.84 | 0.86 | 0.89 |
| | PLBART (3) | 4 | 0.19 | 14.17 | 0.83 | 0.86 |
| {Keyword-Identifier} | CodeBERT (5) | 7 | 0.0 | 6.68 | 0.87 | 0.91 |
| | CodeBERT (9) | 4 | 0.31 | 3.67 | 0.88 | 0.91 |
| | GraphCodeBERT (5) | 4 | 0.18 | 0.81 | 0.87 | 0.92 |
| | GraphCodeBERT (9) | 4 | 0.20 | 4.33 | 0.79 | 0.91 |
| | UniXcoder (5) | 4 | 0.13 | 6.43 | 0.82 | 0.86 |
| | UniXcoder (9) | 3 | 0.11 | 0.72 | 0.76 | 0.83 |
| | CodeT5 (5) | 4 | 0.16 | 7.38 | 0.76 | 0.81 |
| | CodeT5 (9) | 4 | 0.52 | 19.72 | 0.81 | 0.85 |
| | PLBART (3) | 4 | 0.13 | 11.77 | 0.78 | 0.78 |

Table 7: Results of analysis by DirectProbe for data flow edge prediction with 3 labels.

| Tokens | Model (Layer) | No. of clusters | Distance | | Label Accuracy | | |
|---|---|---|---|---|---|---|---|
| | | | Min | Avg | No Edge | ComesFrom | ComputedFrom |
| {Identifier-Identifier} | CodeBERT (5) | 5 | 0.36 | 7.59 | 0.70 | 0.95 | 0.94 |
| | CodeBERT (9) | 5 | 0.42 | 7.54 | 0.70 | 0.95 | 0.94 |
| | GraphCodeBERT (5) | 4 | 0.41 | 2.32 | 0.68 | 0.94 | 0.94 |
| | GraphCodeBERT (9) | 4 | 0.51 | 2.90 | 0.73 | 0.95 | 0.95 |
| | UniXcoder (5) | 4 | 0.41 | 4.89 | 0.66 | 0.93 | 0.91 |
| | UniXcoder (9) | 4 | 0.34 | 4.20 | 0.64 | 0.90 | 0.88 |
| | CodeT5 (5) | 6 | 0.0 | 3.40 | 0.69 | 0.92 | 0.81 |
| | CodeT5 (9) | 4 | 1.57 | 15.00 | 0.63 | 0.90 | 0.91 |
| | PLBART (3) | 6 | 0.0 | 4.76 | 0.68 | 0.90 | 0.83 |

Table 8: Cluster size and label for last layer of models for tree distance prediction task

**CodeBERT**

| Cluster | 0 | 1 | 2 | 3 | 4 | 5 | 6 | 7 | 8 | 9 |
|---|---|---|---|---|---|---|---|---|---|---|
| Label | 3 | 2 | 3 | 5 | 2 | 6 | 6 | 4 | 5 | |
| Size | 178 | 806 | 453 | 225 | 241 | 400 | 683 | 357 | 1042 | 815 |

**GraphCodeBERT**

| Cluster | 0 | 1 | 2 | 3 | 4 | 5 | 6 | 7 | 8 |
|---|---|---|---|---|---|---|---|---|---|
| Label | 2 | 3 | 5 | 3 | 2 | 6 | 5 | 6 | 4 |
| Size | 48 | 386 | 94 | 645 | 999 | 921 | 946 | 119 | 1042 |

**UniXCoder**

| Cluster | 0 | 1 | 2 | 3 | 4 | 5 | 6 | 7 | 8 | 9 | 10 | 11 | 12 |
|---|---|---|---|---|---|---|---|---|---|---|---|---|---|
| Label | 3 | 4 | 6 | 4 | 6 | 3 | 2 | 2 | 5 | 4 | 3 | 5 | 6 |
| Size | 334 | 377 | 225 | 337 | 83 | 168 | 662 | 385 | 646 | 328 | 529 | 394 | 732 |

**CodeT5**

| Cluster | 0 | 1 | 2 | 3 | 4 | 5 | 6 | 7 | 8 | 9 |
|---|---|---|---|---|---|---|---|---|---|---|
| Label | 5 | 2 | 3 | 2 | 3 | 6 | 5 | 4 | 5 | 6 |
| Size | 26 | 653 | 354 | 394 | 677 | 156 | 61 | 1042 | 953 | 884 |

**PLBART**

| Cluster | 0 | 1 | 2 | 3 | 4 | 5 | 6 | 7 | 8 |
|---|---|---|---|---|---|---|---|---|---|
| Label | 2 | 2 | 3 | 3 | 6 | 5 | 6 | 4 | 5 |
| Size | 105 | 942 | 614 | 417 | 227 | 183 | 813 | 1042 | 857 |

Table 9: Cluster size and label for last layer of models for siblings prediction task

| | | 0 | 1 | 2 | 3 | 4 | 5 | 6 |
|---|---|---|---|---|---|---|---|---|
| CodeBERT | Cluster | 0 | 1 | 2 | | | | |
| | Label | Sibling | Sibling | Non-sibling | | | | |
| | Size | 411 | 779 | 1210 | | | | |
| GraphCodeBERT | Cluster | 0 | 1 | 2 | 3 | | | |
| | Label | Sibling | Non-sibling | Non-sibling | Sibling | | | |
| | Size | 1 | 53 | 1157 | 1189 | | | |
| UniXcoder | Cluster | 0 | 1 | 2 | 3 | | | |
| | Label | Non-sibling | Sibling | Non-sibling | Sibling | | | |
| | Size | 2 | 1153 | 1208 | 37 | | | |
| CodeT5 | Cluster | 0 | 1 | 2 | 3 | 4 | 5 | 6 |
| | Label | Sibling | Non-sibling | Non-sibling | Sibling | Sibling | Sibling | Non-sibling |
| | Size | 664 | 458 | 135 | 157 | 365 | 4 | 617 |
| PLBART | Cluster | 0 | 1 | 2 | 3 | 4 | | |
| | Label | Sibling | Sibling | Non-sibling | Sibling | Non-sibling | | |
| | Size | 610 | 126 | 33 | 454 | 1177 | | |

Table 10: Cluster size and label for last layer of models for data flow edge prediction task

| | | 0 | 1 | 2 | 3 | 4 | 5 | 6 |
|---|---|---|---|---|---|---|---|---|
| CodeBERT | Cluster | 0 | 1 | 2 | 3 | | | |
| | Label | NoEdge | NoEdge | Comes | Computed | | | |
| | Size | 1 | 1208 | 1206 | 1185 | | | |
| GraphCodeBERT | Cluster | 0 | 1 | 2 | 3 | 4 | 5 | 6 |
| | Label | Computed | NoEdge | Computed | NoEdge | Computed | NoEdge | Comes |
| | Size | 1 | 1 | 1008 | 549 | 176 | 659 | 1206 |
| UniXcoder | Cluster | 0 | 1 | 2 | 3 | | | |
| | Label | NoEdge | Computed | NoEdge | Comes | | | |
| | Size | 1 | 1185 | 1208 | 1206 | | | |
| CodeT5 | Cluster | 0 | 1 | 2 | 3 | | | |
| | Label | NoEdge | Computed | NoEdge | Comes | | | |
| | Size | 1 | 1185 | 1208 | 1206 | | | |
| PLBART | Cluster | 0 | 1 | 2 | 3 | | | |
| | Label | NoEdge | Computed | NoEdge | Comes | | | |
| | Size | 1 | 1185 | 1208 | 1206 | | | |

