# OpenReview forum: "A Critical Study of What Pre-trained Code Models (do not) Learn"
_ICLR.cc/2024/Conference — Submitted to ICLR 2024_

### Official Review · Reviewer_Yafu · 2023-10-31

**Soundness:** 2 fair
**Presentation:** 2 fair
**Contribution:** 2 fair
**Rating:** 3
**Confidence:** 4

**Summary:**

The paper is a critical study of pre-trained code models, focusing on what kinds of information are or aren’t encoded in the PCMs, to explain why PCMs fail to generalize beyond the datasets they are trained on. The paper creates model graphs by attention analysis and compares them with syntax graphs, dataflow graphs, and non-identifier graphs. Based on the comparison results, the paper argues that while PCMs encode syntactic and data flow relations in the attention layers, but fail to encode relations between syntactic and identifier tokens. The paper further performs a distance prediction task with DirectProbe on hidden representations and finds that hidden representations do not encode enough information to discriminate between different identifier types and syntax structures, which may lead to syntactic errors in the outputs of PCMs.

**Strengths:**

1. A critical study of several popular PCMs with careful analysis of their attention maps and hidden representations to reveal the limitations of those PCMs on encoding specific relations.
2. The paper points out some wrong assumption used by prior works.

**Weaknesses:**

1. To my understanding, most of the studied models are primarily token-based models. It is somewhat unexpected by construction that the models will have accurate knowledge of AST and DFG. Also, depending on the tasks it may not matter very accurate knowledge of these code properties. For example, many previous studies have shown that for code summarization tasks, a good choice of variable names is good enough. It is not very clear from the paper what kind of properties primary token-based models are expected to learn and how they have learned it.

2. Why do authors expect the self-attention score between syntactic and identifier nodes to be high?

3.  Overall, the paper's presentation is poor. It does not have a clear and logical structure, which poses difficulty in reading and understanding. The visualization is not clear. Explanations of the figures and data are not detailed enough, so it is a bit hard to relate them to the conclusions.

4. While the use of the “motif structure” seems integral to the construction of the code graph, it is not explicitly defined in the work; providing some background or context for this concept may be critical.

5. The paper makes some unsubstantial claims. For example, in the intro, they say one group of papers says LLMs are working for code, while some other group of papers find cases where LLMs fail. The authors say these two trends are contradictory. I think this is how any field progresses ---they are not necessarily contradictory trends.

**Questions:**

1. Why do you set all the values above threshold to 1? Why not maintain an individual attention score?

2. What is the intuition behind expecting a higher self-attention score between identifiers and keywords, especially for a token-cased model?

---

> ### Author Response · Authors · 2023-11-22
>
> We thank the reviewer for the positive comments about the papers and for pointing out the weaknesses. Based on the suggestion, we have made changes to the paper which we enumerate in the common response to all reviewers. We kindly ask the reviewer to go through those comments. In the following, we address the weaknesses pointed out and the questions raised by the reviewer.
>
> **Weaknesses**
> 1. These models indeed act on tokens but they do not just take in some tokens and split out a new one. They are optimized to perform some task. There is an implicit assumption that structure in input data can be learned by the models from simple inputs with proper optimizations. Evidence has been provided to support this assumption by multiple studies in different fields, including NLP and vision. Previous work in NLP has shown that with tokens as inputs and proper optimization, models can learn the language grammar. Similarly, code models have been shown to learn program syntax. Our work is an attempt to understand how much structure can be encoded in models with optimizations based on tokens. So, we believe it is not correct to say it should be unexpected using construction.
>
>     Our work points out a significant limitation in learning these structures about code, which has been previously unexplored. We do not claim that the learning should be perfect; we only point out that some very important relations necessary for comprehending code syntax and code logic are not encoded by the models in the self-attention values and hidden representations.
>
>     Moreover, not all models are trained only with tokens. GraphCodeBERT takes data flow graph nodes as inputs along with tokens, while, UniXcoder takes nodes of flattened AST as input.
>
>     The reviewer rightly points out that for code summarization a good choice of variable names is good enough. However, we argue that such dependence on function names as opposed to code logic for summarization is a symptom of shortcut learning and hinders generalization. We have added a discussion on this argument in Section 4.3 of the updated paper as well as in the common comments to all reviewers. We kindly ask the reviewer to consider these too.
>
> 2. We assume self-attention between syntactic and identifier tokens to be high only if they are within the same motif structure. This assumption is based on results in previous works such as Wan et al (2022). Further, the relation between syntactic tokens and identifiers plays an important role in program flow. For example, `if` and `else` defines code blocks but the flow of the program is decided by the identifiers related to these syntactic tokens.
> 3. We have made changes to the paper described in general comments to make the flow and the conclusions of the paper more clear. We have also modified the captions of Figures and Tables to add additional details.
> 4. The term 'motif structure' was defined in Wan et al (2022) and we use the definition as is. The citation was missing in the Introduction section. Following the suggestion of the reviewer, we have described the motif structure in Appendix B of the updated paper and have cited the relevant work in the Introduction.
> 5. Based on the suggestions of the reviewer, we have made changes to the Introduction section to make our claims clearer and have also reworded the text about contradictory trends.
>
> **Questions**
> 1. We had initially set the values to 1 to replicate previous work and then continued our analysis with this value to ensure our work remained comparable to those works. We can use the individual attention values by weighing all computations with the original value. Since the attention values are between 0 and 1, it will make the limitations of the models more stark. But, it will not change the conclusions drawn from the analysis. Setting it to 1 shows that the limitations remain even if the attention is very high. We have added this explanation in Section 3.2.1 of the updated paper.
> 2. Please see point 2 of weakness.

---

> ### Comment · Reviewer_Yafu · 2023-11-22
>
> Thanks for your detailed comments and update the paper. While I agree token based models should understand some approximate structures, aligning the models to different downstream tasks may enforce them to learn the structures better. However, as you have conducted the experiments with a pre-trained model, where most of the models are not challenged to learn the structure, it is not clear to me why we would expect the model to learn the structure.
>
> Authors claim the models should learn some approximate structure----in such case, there should be some estimate for approximation.
>
> I think with more experiments with more downstream tasks may clarify some of these points better.

---

> > ### Author Response · Authors · 2023-11-23
> >
> > Thank you for bringing up this important point. We would like to mention that all models that we analyzed, with the exception of PLBART, were pre-trained on code-based objectives. So, they are challenged to learn code structures. For instance, CodeBERT was trained using Replaced Token Detection (RTD) along with Masked Language Modeling. Suppose a variable, `v`, is replaced with another variable, `v’`. To determine if `v’` is the original variable or not, the model needs to understand how the data flows through the code. Other models use more complex code-based objectives. For example, CodeT5 is trained with bimodal dual generation and identifier-aware pre-training objectives.
> >
> > We still have an important question, however. Does training these models on downstream tasks help them learn more structures than just pre-training? While it may seem intuitive, in practice, this isn't always the case. For example, [1] studied the effect of fine-tuning on PCMs. They observed information loss between pre-trained and fine-tuned models on 2 discriminative and 3 generative tasks. The study also found that models trained from scratch on a downstream task encoded even less information.
> >
> > [1] "Probing Pretrained Models of Source Codes", Troshin et al., BlackboxNLP Workshop on Analyzing and Interpreting Neural Networks for NLP, 2022

---

### Official Review · Reviewer_vS1o · 2023-10-31

**Soundness:** 3 good
**Presentation:** 3 good
**Contribution:** 3 good
**Rating:** 5
**Confidence:** 3

**Summary:**

This paper presented an analysis of the self-attention mechanism and the hidden representations within PCMs (pre-trained code models). The study reveals that while PCMs do encode syntactic and data flow relations in self-attention, they only encode relations within specific subsets of input tokens and do not encode syntactic-identifier relations between code tokens. The authors believe that this limitation is what leads to syntactic errors in outputs of PCMs. The authors also observe that this problem persists across different model architectures, datasets, and pre-training objectives.

**Strengths:**

- The paper targets an important problem.
- Detailed analysis. It is commendable that the authors performed a visualization analysis of the hidden states of pre-trained code models.

**Weaknesses:**

This paper targets an important problem of understanding why PCMs work/don’t work. The findings could provide useful insights that could help address current models’ limitations and inspire further research in this area.

Currently, the analysis was performed on 5 PCMs that were proposed in 2020-2022 and are relatively small. No investigation of more recent models, especially LLMs (such as CodeLLaMa and Starcoder), is performed. It is not clear if the obtained findings are applicable to more recent, large models.

It seems that this work closely aligns with the research conducted by Wan et al. (2022) and places a greater emphasis on the fine-grained details of code tokens, achieved by categorizing input tokens into syntactic tokens and identifiers. From this standpoint, this work is considered a bit incremental.

The authors claimed that their findings shed light on why PCMs fail to generalize beyond dataset they are trained on and in real world applications. I do not think this statement can be well supported by the findings shown in this paper. The authors investigate the inability of pre-trained code models to comprehending relations across the syntactical keywords and identifier tokens, however, the connections between the comprehending ability and the generalization ability are non-trivial and need to be explained explicitly.

Several statements in this paper are not consistent. For example, the research objectives are inconsistent, the study attempts to explore the extrapolation ability of a model in the introduction, however, the methods mainly focus on evaluating the ability of a model distinguishing syntactical tokens and identifiers.

In the Introduction, the authors point out that certain prior works often make incorrect assumptions in their experimental settings. Unfortunately, the authors do not specify what kinds of incorrect assumptions these works often make.

**Questions:**

- Are the obtained findings applicable to more recent, large models?

- What are the connections between the comprehending ability and the generalization ability?

---

> ### Author Response · Authors · 2023-11-22
>
> We thank the reviewer for the positive comments about the papers and for pointing out the weaknesses. Based on the suggestion, we have made changes to the paper which we enumerate in the common response to all reviewers. We kindly ask the reviewer to go through those comments. In the following, we address the weaknesses pointed out and the questions raised by the reviewer.
>
> **Weaknesses**
> 1. Selection of models: We agree that the paper focuses on slightly older and smaller models. However, we believe that the analysis of these models is still relevant and provides important insights for future research. We provide our argument for selecting these models in common response to all reviewers and also in Appendix C of the updated paper.
> 2. Incremental work: We disagree with this comment. While the initial motivation was to replicate the work by Wan et al. (2022), we found that the results and conclusions of that work were significantly affected by the choice of assumptions that were made. Subsequently, we did a comprehensive analysis of existing works in this domain and found that such assumptions abound.  We, then, extended previous works to point out the limitations of current models. We used DirectProbe, which has not been used for PCMs before. For analysis with DirectProbe, we did a complete experimental design and created a dataset to run DirectProbe. We also extended the existing analysis to more models and data flow graphs. In previous work, data flow graphs were only used with probing-based classifiers with an assumption of linearity in encoding. Our work revealed that the linearity assumption is not valid. Similarly, our work is the first to use graph edit distance for attention analysis, which provided new insights into the limitations of PCMs.
> 3. We agree with the reviewer that the original paper did not discuss how our findings shed light on why PCMs fail to generalize beyond dataset. We have now added details explaining the relation between the ability of models to comprehend code logic and syntax and their generalization ability in Section 4.3 of the update paper.
> 4. The research objective of the paper is to understand the limitations of PCMs with respect to what code information they do not encode. We have stated this in the Abstract as well as in the Introduction. We had additionally claimed that the limitations we found shed light on the limitations of models to generalize. However, we had not made this connection clear in the paper. We now do so in Section 4.3 of the updated paper.
> 5. In the last paragraph of the Introduction, we have detailed the non-systematic assumptions made by previous works. These are the choice of attention threshold and evaluation metric for attention analysis as well as the assumption of linearity in information encoded in hidden representation. In our analysis, we found that these assumptions impact the conclusions drawn from the analysis (See Section 3.2.1 for attention analysis and Section 4.2 for linearity assumption) and can lead to wrong conclusions.
>
> **Questions**
> 1. This cannot be answered without evaluating the larger models. However, a comparison between PLBART and CodeT5, two encoder-decoder models with different sizes, did not reveal any difference in the model's ability to understand syntax better only because of size. Moreover, as we explain in Appendix C of the updated paper, analysis of smaller models also reveals relevant insights.
> 2. We now address this question in the last two paragraphs of Section 4.3 of the updated paper and have also explained in the general comments for all reviewers.

---

### Official Review · Reviewer_GkFF · 2023-11-01

**Soundness:** 3 good
**Presentation:** 3 good
**Contribution:** 3 good
**Rating:** 5
**Confidence:** 4

**Summary:**

This paper presents a critical study of pre-trained code models (PCMs) to understand what they do and do not learn regarding code relations. By analyzing the self-attention mechanism and hidden representations, the authors reveal that while PCMs encode syntactic and data flow relations among input tokens, they fail to encode relations between syntactic tokens and identifiers. This limitation results in hidden representations not being able to discriminate between different identifier types and syntax structures. The authors show that these learning gaps persist across different model architectures, datasets, and pre-training objectives, providing insights into why PCMs fail to generalize beyond the dataset they are trained on and in real-world applications. The findings encourage further research to address the limitations of current PCMs and develop more robust experimental designs for model interpretability and improvement in training methods.

**Strengths:**

The paper exhibits originality by providing a fine-grained analysis of pre-trained code models, uncovering previously unaddressed limitations. The quality is high due to its critical examination of assumptions in prior work, leading to more accurate conclusions. The paper's clarity is evident in its well-written presentation, making the experiments and findings easily understandable. Its significance lies in inspiring further research to address the identified limitations and develop more robust experimental designs and training methods for PCMs.

**Weaknesses:**

1. The scope of the models and datasets analyzed is limited, and the paper could be strengthened by considering a wider range of model architectures, sizes, and training data sources to provide a more comprehensive understanding of the limitations across various PCMs.
2. The paper focuses primarily on syntax and data flow relations but could extend its analysis to other aspects, such as the influence of natural language on code understanding and the alignment between natural language and programming languages.
3. The experiments conducted in the paper focus on Python code. Including other programming languages in the analysis would provide insights into whether the observed limitations are language-specific or general across different programming languages.

**Questions:**

1. How do the observed limitations in PCMs affect their performance on real-world tasks? It would be helpful to provide examples or case studies to illustrate the practical implications of these limitations.
2. Are the identified limitations specific to the Transformer-based models studied in the paper, or do they extend to other types of models, such as RNNs or LSTMs, in the context of code understanding?
3. Have the authors considered exploring the impact of different pre-training objectives or methods that could potentially address the limitations identified in the paper? Some suggestions or proposals to improve the training process would be valuable.
4. How do the limitations in the PCMs affect their ability to learn from more diverse training data, such as codes from different domains or programming languages? Would incorporating such variety during the training phase help in mitigating the observed limitations?

---

> ### Author Response · Authors · 2023-11-22
>
> We thank the reviewer for the positive comments about the paper and for pointing out the weaknesses. Based on the suggestion, we have made changes to the paper which we enumerate in the common response to all reviewers. We kindly ask the reviewer to go through those comments. In the following, we address the weaknesses pointed out and the questions raised by the reviewer.
>
> **Weaknesses**
> 1. Limited choice of model architectures, sizes, and data sources:
>
>      We have explained the limitations regarding model architecture in Appendix C of the updated paper and also mentioned it in common response to all reviewers.
>
>     The limited choice of the dataset was deliberate to remove any effect of data distribution shift. All models except PLBART were trained on the CodeSearchNet (CSN) dataset. Also, CSN is a fairly large dataset with 6 programming languages and 2 million comment-code pairs. PLBART was trained on Java and Python code from Google BigQuery. Despite a different dataset and the potential for distribution shift, we considered PLBART due to its smaller size compared to other encoder-decoder models.
>
>     PLBART is about half the size of encoder-decoder models such as CodeT5, but the difference in sizes does not seem to affect the conclusion. Moreover, the idea behind analyzing smaller models was to understand the limitations of smaller models, to improve them instead of always going for larger sizes. We have added a discussion on the use of smaller models in Appendix C of the updated paper.
>
> 2. Alignment between Natural and Programming Language:
>
>     We agree that understanding the effect of natural language and NL-PL alignment is important and see it as the next logical extension of our work. However, we consider this to be out-of-scope for this paper because it is non-trivial to extend the current work to natural languages. The major challenge with natural language is that semantically similar texts can lead to significantly different output, as is seen with various prompting strategies and in-context learning. Thus, we cannot simply take a specific text-code pair and analyze the models. Instead, we need to consider which version of the text can give the best output and which one gives the poorest. To do this, we are currently working towards a statistical method to analyze NL-PL alignment in a principled manner. Moreover, analyzing only on code is also important for many downstream tasks such as code completion, clone detection, code retrieval etc.
> 3. Python-only Analysis:
>
>     We agree that including more languages in the analysis would provide additional insights. However, the analysis of Python in itself is very relevant and important. PCMs perform significantly better on Python compared to other languages. Better performance on Python has also resulted in works such as Math Coder which generates Python code to solve math problems. In general, Python has become the primary focus of recent works. Moreover, widely used benchmarks such as HumanEval, MBPP and DS1000 have reference solutions in Python and so, the recent state-of-the-art models are evaluated usually on Python. Similarly, the models analyzed in the paper perform significantly better on Python benchmarks compared to other programming languages.

---

> > ### Author Response · Authors · 2023-11-22
> >
> > **Questions**
> > 1. Thank you for this suggestion. We have updated the section on the limitations of PCMs (Section 4.3 in the updated paper) to discuss how the limitations affect PCMs on real-world tasks and in generalization beyond training data. We have also added these explanations in common response to all reviewers.
> > 2. We cannot comment on whether these limitations extend to RNNs / LSTMs as well without analyzing code models with RNN / LSTM architecture. Our focus in this paper was on models based on transformer architecture. We agree with the reviewer that extending this work to RNNs / LSTMs would be very interesting.
> > Unfortunately, there are very few works on code models which use RNNs/LSTMs with attention and even fewer with self-attention. More importantly, though, the way self-attention works in RNNs is different from that of transformers. The attention in RNN is over hidden representation at different time steps. Hidden state H_t has seen all tokens till t, and so is not independent for each token. Moreover, the self-attention matrix is not always the same size as the input. For example, [3] has an attention matrix of size r * n for an input of n tokens. Our work assumes an n * n attention matrix, which is valid for transformer architecture, and so cannot be directly extended to RNNs / LSTMs. Finally, the existing RNN / LSTM models are trained on much smaller datasets.
> > This said, inspired by this helpful suggestion, in future, we will look at modifying self-attention-based LSTM architecture to have n * n attention matrix or train recurrence-based transformer models such as [4] on code and analyze and compare them with transformers. This would require modifying some models, training them with code datasets such as CSN and with different inputs and objectives. We feel this would be a research in itself and should be considered a topic of another paper.
> > 3. We did consider the impact of different pre-training objectives and had made suggestions in Section 4.3 of the original paper. We discuss these suggestions in Section 4.4 of the updated paper and have made changes to the text so that the impact of pre-training objectives and the suggestions are hopefully stated more clearly.
> > 4. This is a very important point, which is already considered in the paper. All the models studied in the paper have been trained on multiple languages. PLBART has been trained on Python and Java, while the others have been trained on the 6 languages present in the CSN dataset. Moreover, CSN is a huge dataset with 2 million comment-code pairs. Thus, the limitations exist despite the variety in training data. We have modified Appendix E of the paper to include these details.
> >
> > [3] "A structured self-attentive sentence embedding", Lin et al., ICLR 2017
> >
> > [4] "Encoding recurrence into Transformers", Huang et al., ICLR 2023

---

### Official Review · Reviewer_gkoc · 2023-11-05

**Soundness:** 3 good
**Presentation:** 2 fair
**Contribution:** 2 fair
**Rating:** 5
**Confidence:** 3

**Summary:**

The paper analyzes the limitations of pre-trained code models (PCMs) by studying what relations they do and do not encode in their internal representations. It focuses on syntactic, data flow, and semantic relations between code tokens.

The main finding is that while PCMs encode some syntactic and data flow relations, they fail to encode relations between syntactic tokens like keywords and identifiers like variable names.

The authors perform comprehensive attention analysis and probing of hidden representations across multiple models like CodeBERT, GraphCodeBERT, CodeT5, etc. The limitations are found to persist across models, architectures, objectives and datasets.

**Strengths:**

1. **Comprehensive analysis of multiple relation types.** A key contribution is analyzing syntactic-syntactic, identifier-identifier, and syntactic-identifier relations separately. This allows the authors to uncover that while models encode some relations well, they fail to encode syntactic-identifier relations. The fine-grained categorization and analysis provides valuable insights into exactly where models fall short.

2. **Rigorous experimental methodology.** The paper investigates limitations rigorously through both attention analysis and probing of hidden representations. The authors also avoid common pitfalls by using suitable thresholds, metrics, and probing techniques. The robust experimental design lends credibility to the conclusions drawn.

3. **Analysis across multiple models and settings.** Rather than evaluating a single model, the study analyzes limitations across transformer architectures, training objectives, and datasets. The consistency of observations across these varying settings strongly indicates fundamental, widespread limitations in encoding certain code relations. The cross-model analysis strengthens the paper's central claim regarding limitations of pre-trained code models.

**Weaknesses:**

1. The selection of models analyzed is limited. While the paper has covered 5 models that are either encoder-only or encoder-decoder, it has not covered decoder-only PCMs, which seem to be the major choice of pre-trained language models currently.
2. Lack of real-world evaluation. The paper focuses on analyzing model internals. It does not evaluate how the limitations discovered actually affect model performance on downstream tasks. Testing on real-world code generation and code search benchmarks could better highlight the practical implications.
3. The paper does not provide architecture-related or data-related explanation for the limitations of current PCMs. Also, it does not propose methods to address the limitations identified. Providing ideas to improve relation encoding could make the work more constructive.

**Questions:**

Please see the weakness section.

---

> ### Author Response · Authors · 2023-11-22
>
> We thank the reviewer for the positive comments about the paper and for pointing out the weaknesses. Based on the suggestion we have made changes to the paper which we enumerate in the general response to all reviewers. We kindly ask the reviewer to go through those comments. In the following, we address the weaknesses pointed out by the reviewer.
> 1. On limited selection of models:
> Since this weakness was mentioned by multiple reviewers, we have explained how the models we have analyzed provide important and relevant insights in the common response to all reviewers and also in Appendix C.
>  2. Lack of real-world evaluations:
> Since this weakness was mentioned by multiple reviewers, we have provided explanations in common response to all reviewers. Based on the suggestion, in Section 4.3 of the updated paper, we discuss how our work relates to real-world tasks and limits to generalization.
> 3. Suggestions to address limitations:
> Based on our analysis, we provided some suggestions in Section 4.3 of the original paper. However, the suggestions were not clear. In the updated paper, we discuss these suggestions in Section 4.4 after pointing out the limitations. Further, we have modified the section to make the suggestion more clear.

---

> > ### Comment · Reviewer_gkoc · 2023-11-22
> >
> > Thank you for the response.
> >
> > While it has solved some of my concerns, I find the discussion without experimental evidence not convincing enough. Hence I'm not changing my rating.

---

### Author Response · Authors · 2023-11-22
**Common response to all reviewers**

Thank you for the comments and suggestions. Based on suggestions, we have made the following changes to the paper:
1. We have updated the text of the Introduction to make it clearer and easier to understand. We have also made the claims of the paper clearer. The general details in the Introduction remain the same as in the original paper.
2. In Section 3.2.1, we have made changes to explain the reasons for setting self-attention values to 1 before analysis.
3. Section 4.3 (The Effect of Input and Pre-training Objectives) had suggestions that could alleviate current limitations. Based on the comments, we feel the suggestions were not clear enough. We have moved this section below limitations and have further modified it to make the suggestions more clear.
4. In the original paper, we had not discussed how our work sheds light on limitations to generalizations and effects on real-world tasks. To address this, we have added additional discussion to the section "Limitations of PCMs".
5. To make space for the additional details, we have moved parts (AST, DFG and transformer model) of Section 2 (Background Details) to Appendix B. In Appendix B, we also explain motif structures used in our analysis.
6. We have added a new section to the Appendix (Appendix C), which compares encoder-only, encoder-decoder and decoder-only models in more detail and points out why the first two architectures are still relevant despite the preference for decoder-only models in recent works.
7. We have modified captions of Figures and Tables to make them more informative.

Since a limited selection of models and effects on real-world tasks were concerns of almost all reviewers, we provide an explanation here (along with adding these details to the paper).

**On limited selection of models**:
While we agree that decoder-only models could additionally be included in the analysis and will provide additional insights, we believe that the analysis of smaller encoder-only and encoder-decoder models already provides insights which are significant and potentially relevant for future work. Smaller encoder-only and encoder-decoder models remain relevant for many tasks such as code retrieval, detection tasks etc. The application of decoder-only models remain limited to generation tasks (and fill-in-the-middle task in some recent models).  For example, The StarCoder work used a 125M parameter encoder-only model to detect and remove personally identifiable information in the code dataset before training StarCoder. Even on the generation task, small encoder-decoder models can match the performance of much larger decoder-only models. For example, encoder-decoder model CodeT5+ with 220M parameter matches the performance of decoder-only CodeGen-mono with 2B parameters on MathQA and GSM8K benchmarks.

Also, our choice of models encompasses significant variations in input and pre-training objectives. For instance, PLBART is trained only with de-noising objectives.  CodeBERT uses the commonly used MLM objective from NLP and code-based replaced token detection. GraphCodeBERT extended CodeBERT to include data flow in input and data flow-based objectives, CodeT5 used identifier-aware pre-training and UniXcoder used AST in input along with contrastive learning-based pre-training.

Through this analysis of different inputs and objectives, we were able to find that including code property-based objectives in training, as with GraphCodeBERT and CodeT5, helps in learning code properties better (explained in Section 4.4 of the updated paper). Due to the uni-directional/ non-causal training of decoder-only models, code properties such as AST and data flow relations have not been included in the training objectives of decoder-only models.

Moreover, the purpose of an encoder is to build a representation of the input, while a decoder is optimized for generation. Building a good low-dimensional representation of input data is important for efficient learning. This is missing from a decoder-only model. While decoder-only models work well despite this limitation, the gains in performance are only observed at a very high scale. Well-performing decoder-only models have more than a billion parameters. While the availability of compute and huge datasets have made training these models easy, the environmental costs cannot be ignored. Also, the increasing size could lead to models memorizing more data, instead of learning which has been documented in a few previous works (mentioned in Related Works in paper).

Finally, using smaller models not only provides environmental benefits but also ensures that teams with limited compute resources can also experiment and contribute to the field.

---

> ### Author Response · Authors · 2023-11-22
> **Common response to all reviewers (contd.)**
>
> **Performance on real-world tasks and limits to generalization:**
> While it is true that we have not evaluated the models on real-world tasks in the paper, previous works have contributed and documented such evaluations. Our work is motivated by these previous works. Most prominently, we point to GraphCodeBERT which brings out mistakes such as random identifiers and CodeRL which also evaluates CodeT5 for syntactic mistakes.
>
> The papers of all the models we analyzed have also provided details of evaluation on various benchmarks. While models have improved significantly on benchmarks, they still fail to generalize to real-world tasks. Furthermore, the evaluation on some downstream tasks, such as code summarization, also highlights another dimension to limitations –  learning shortcuts. Studies such as [1] have shown that for generating correct summaries PCMs often rely on function names and code comments, instead of code logic. The tendency of neural networks to learn shortcuts has been well documented [2].
>
> In section 4.3, we show that models fail to encode sufficient information about code to comprehend subtle syntactic differences and code logic. Without proper comprehension of code, PCMs have to rely on shortcut cues for prediction. Thus, shortcuts in PCMs arise due to their inability to encode relevant code information. Since these shortcuts may not be available outside of training data, PCMs fail to generalize. For instance, models can fail when developers do not use proper function names, they do not comment their code properly, or comments and function names of real code differ from the patterns seen by the model during training.
>
> [1] "Code summarization: Do transformers really understand code?", Sontakke et al., Deep Learning for Code Workshop, 2022
>
> [2] "Shortcut learning in deep neural networks", Geirhos et al., arxiv, 2020

---

### Meta-Review · Area_Chair_71gk · 2023-12-01

**Metareview:**

This paper studies pre-trained code neural models aiming to measure what information is or isn’t encoded. The core finding is that while these models encode some syntactic and data flow relations, they fail to encode relations between syntactic tokens like keywords and identifiers like variable names.


### Strengths
- Understanding the “internals” of Transformers for coding applications is an important task that can lead to useful insights for improving them and their training methods.

### Weaknesses
- While pretrained models form the base of most modern code neural models, they are not tied to a downstream task. This bears the question if the presented results would still hold if these models would be fine-tuned for a particular downstream task that requires some understanding of code’s data/control flow, etc. In that sense, it’s also unclear if the observations are due to the pretraining objective or the transformer architecture.
- It is unclear how the scale of these models (num layers, hidden size) affect the observed results. Are some of the issues detected resolved with scale or not?

Given the above, I recommend for this paper to not be accepted at its current state. Having said that, I would encourage the authors to address these concerns and resubmit.

**Justification For Why Not Higher Score:**

The value of the work at its current state is unclear.

**Justification For Why Not Lower Score:**

n/a

---

### Decision · Program_Chairs · 2024-01-16

Reject